

# Sparse 3D reflection seismic survey for deep-targeting iron-oxide deposits and their host rocks, Ludvika Mines-Sweden

Alireza Malehmir[1], Magdalena Markovic[1], Paul Marsden[2], Alba Gil [1], Stefan Buske[3], Lukasz Sito[4], Emma Bäckström[2], Martiya Sadeghi[5] and Stefan Luth[5]

[1]Department of Earth Sciences, Uppsala University, SE 75236, Uppsala, Sweden
[2]Nordic Iron Ore AB, Sweden
[3]TU Bergakademie Freiberg, Germany
[4]Geopartner, Poland
[5] Geological Survey of Sweden, Uppsala, Sweden

*Correspondence to*: Alireza Malehmir (alireza.malehmir@geo.uu.se)

**Abstract.** Many metallic mineral deposits have sufficient contrasts, particularly density, to be detectable using seismic methods. These deposits are sometimes significant for our society, economic growth and can help to accelerate the energy transition towards decarbonization. However, their exploration at depth requires high-resolution and sensitive methods. Following a series of 2D seismic trials, a sparse, narrow source-receiver azimuth, 3D seismic survey was conducted in the

Blötberget mine, in central Sweden, covering an area of approximately 6 km $^2$ for deep targeting iron-oxide deposits and their host rock structures. The survey benefited from a collaborative work by putting together 1266 seismic recorders and a 32t vibrator generating 1056 shot points in a fixed geometry setup. A linear sweep ranging from 10-160 Hz and 20 s long was generated three times per shot point. Shots were fired at every 10 m where possible and receivers placed at every 10-20 m. Notable quality data were acquired although the area is dominated by swampy places as well as by built-up roads and historical

tailings. The data processing had to overcome these challenges in particular for the static corrections and strong surface-waves. A tailored for hardrock-setting-processing workflow was developed for handling such a dataset, where the use of mixed 2D and 3D refraction static corrections were relevant. The resulting seismic volume is rich in terms of reflectivity with clear southeast dipping reflections originated from iron-oxide deposits extending vertically and laterally at least 300 m beyond what was known from boreholes. We estimate potential additional resources from the 3D reflection seismic experiment on the order

of 10 Mt worth drilling for detailed assessments. The mineralization is crosscut by at least two major sets of northwest dipping reflections interpreted to be dominantly normal faults and responsible for much of the lowland in the Blötberget area. Moreover, these post-mineralization faults likely control the current 3D geometry of the deposits. Curved and submerged reflections interpreted from folds or later intrusions are also observed showing the geological complexity of the study area. The seismic survey also delineates the near-surface expression of a historical tailing as a by-product of refraction static

corrections demonstrating why 3D seismic data. The sparse 3D survey illustrates that performing cost-effective reflection surveys for mineral exploration is achievable if they are conducted and planned carefully, systematically and based on earlier experiences.





## 1 Introduction

Mineral exploration industry is challenged to provide fresh resources of the so-called critical raw materials that are important
for green technologies and help accelerate the energy transition towards decarbonization. These critical materials (e.g., rare earth elements or REEs) are often found as associated minerals in other deposits. Hence apart from their own significance, other deposits such as ferrous and non-ferrous need to be found, and the presence and quantity of the critical minerals that are usually beyond the detection limit of geophysical methods studied. Iron-oxide deposits are not an exception; they may contain apatite and a reasonable amount of REEs, Titanium and vanadium. In fact, in terms of tonnage and economic potential, iron
ores are still the number one commodity being mined and consumed worldwide (Fizaine, 2018). However, what makes nowadays discovery of these deposits difficult is their presence at depths because it is generally believed that most economically viable deposits to mine at shallow depths have already been found and exploited and bigger deposits are likely only to be found at depth. Therefore, deep direct targeting requires sensitive high-resolution methods as well as a multidisciplinary approach to avoid an expensive deep drilling exploration program to fail. Lessons and successful progress in
the hydrocarbon industry may be used for deep exploration in hardrock settings although having much more complex geology than those of hydrocarbon settings. Seismic methods, particularly reflection seismics, after being tested now for over three decades in crystalline rock settings (Reed, 1993; Eaton et al., 2003a,b and references therein; Milkereit et al., 2000; Pretorius et al., 2003; Malehmir et al., 2011 and 2012a and references therein; Dehghannejad et al., 2012; Heinonen et al., 2013; Buske et al., 2015 and references therein; Koivisto et al., 2015) are opening their ways into the mineral exploration toolbox as a
standard method. A recent number of publications (Bellefleur et al., 2019 and references therein; Malehmir et al., 2020 and references therein) illustrates why the method is so attractive for deep targeting and mineral exploration.

Nonetheless, most hardrock seismic surveys are conducted either in 2D or in rare cases 3D of various rectangular, using overlapping patches, setups (Adam et al., 2003; Schmelzbach et al., 2007; Malinowski et al., 2012; Manzi et al., 2012, and 2020; Malehmir and Bellefleur, 2009; Malehmir et al., 2012b; Urosevic et al., 2012; White et al., 2012; Cheraghi et al., 2012;
Bellefleur et al., 2015; Maries et al., 2020). Conventional 3D surveys using a high fold and many parallel shot and receiver lines although they satisfy regular sampling (Vermeer, 1998), it requires extensive line cutting and clearance in northern countries. Therefore, they are considered strongly non-environmentally friendly, moreover substantially expensive to conduct. Sparse and dedicated 3D surveys are an alternative to be developed (Bouska 1997; Singh et al., 2019) in which deep targeting is designed so that a particular target will be imaged using a suitable illumination angle.

The Blötberget mining area (Fig. 1) in central Sweden within the so-called Ludvika Mines was the target of an experimental sparse 3D reflection survey given a wealth of several earlier 2D seismic lines (e.g., Malehmir et al., 2017a; Balentrini et al., 2020; Bräunig et al., 2020; Maries et al., 2020; Markovic et al., 2020; Papadopoulou et al., 2020), downhole logging data (Maries et al., 2017) and publicly available high-resolution aeromagnetic data in the area. The seismic survey had two main objectives: (1) to delineate depth and lateral continuation of iron-oxide deposits and (2) to unravel important structures that
may be relevant for deep mining and how the deposits are currently configured and extended at depth. However, the survey



was challenged because of the limited number of receivers available but also logistical challenges including swampy places, canals, dense forests, built-up roads from previous mining activities in the region and historical tailings. The extent or severity of some of these features were only recognized after the survey was completed and during the processing work, as it will be discussed later.

Given that the dataset is currently the subject of a few other studies, the purpose of this inventory publication is how the survey was planned and executed, the standard processing workflow, results and our interpretation of major features in the seismic volume. We demonstrate how a dedicated planning work using a limited number of receivers in an area of approximately 3 by 2 km helped not only to image both vertical and lateral extent of the mineralization but also a number of structural features that crosscut the mineralization. The 3D survey also clearly maps historical tailings in the area as a by-product of the near-

surface static solution. While there have been numerous hardrock seismic surveys conducted in Sweden, this is the first one reported for mineral exploration and should, therefore, be considered as a pilot study encouraging the numerous mining companies in the country to try the method more extensively than only a limited 2D surveys.

## 2 Geology of the study area

Blötberget iron-oxide deposits of the Ludvika Mines sit within one of the three major mineral districts of Sweden known as

Bergslagen (Ripa et al., 2001; Stephens et al., 2009). Bergslagen mineral endowment is diverse from iron oxides to massive sulphides as well as skarns and potentially a good amount of REEs. Iron-oxide deposits are however more known because of their historical importance and for being the cornerstone of the Swedish industry. These deposits are currently not mined. Mining iron-oxide deposits was on a high peak the late 70s (Magnusson, 1970). Then, due to the decrease in the iron ore prices and drop in tonnage, mining these deposits was no longer economically viable especially using underground mining methods

(such as block-caving). Mining has a long history also in Blötberget (our study area). Deposits were mined for 25-30 years until it stopped in 1979 and much of it took place down to 280-360 m depth levels at two of the main deposits. Nordic Iron Ore, who operates the site, plans to restart mining operations at 400-420 m depth level. The company expects to utilize the existing underground infrastructures after the planned restart of the operation and after the necessary renovations and additions. Current mining in the Bergslagen district is mainly focused on massive sulphides in three major underground mines:

Zinkgruvan, Garpenberg and Lovisagruvan (Fig. 1).

In the Ludvika region, however, good quality iron oxides such as Blötberget and Grängesberg have attracted recent attentions. These deposits have high quality magnetite and hematite with 25-60% Fe content but readily upgradable to as high as 71% Fe content with low impurities (e.g., S, Hg, and P) making them highly attractive for iron ore mining and production. The deposits occur within inliers of ca. 1.90-1.85 Ga felsic volcanic rocks (usually metamorphosed) surrounded by migmatite and later

granitic and pegmatitic intrusions (Kathol et al., 2020). Mafic lenses also sometimes occur within the volcanic rocks. Mineralization, in sheet-like occurrences, is usually magnetite dominant but sometimes with some hematite content and some





amount of apatite. In Blötberget, the iron-oxide deposits occur in four sheet-like bodies: Kalvgruvan (apatite-rich magnetite mineralization, 92% magnetite versus 8% hematite), Hugget (apatite-rich magnetite-hematite mineralization, 39% magnetite versus 61% hematite), Flygruvan (apatite-rich magnetite-hematite mineralization, 67% magnetite versus 33% hematite) and

Sandellmalmen (apatite-rich magnetite mineralization). Stratigraphically, the hematite richer zones (Hugget-Flygruvan) overlie the magnetite-rich zones (i.e., Kalvgruvan). According to Jonsson et al. (2013), Grängesberg, which is approximately 10 km southwest of Blötberget and occurs along a similar geological environment with similar magnetic trend, is of Kiruna-type magmatic or high temperature hydrothermal origin. Kiruna is a world-class iron-oxide mine (+600 Mt of proven reserves) producing approximately 3-5% of the world iron ores.

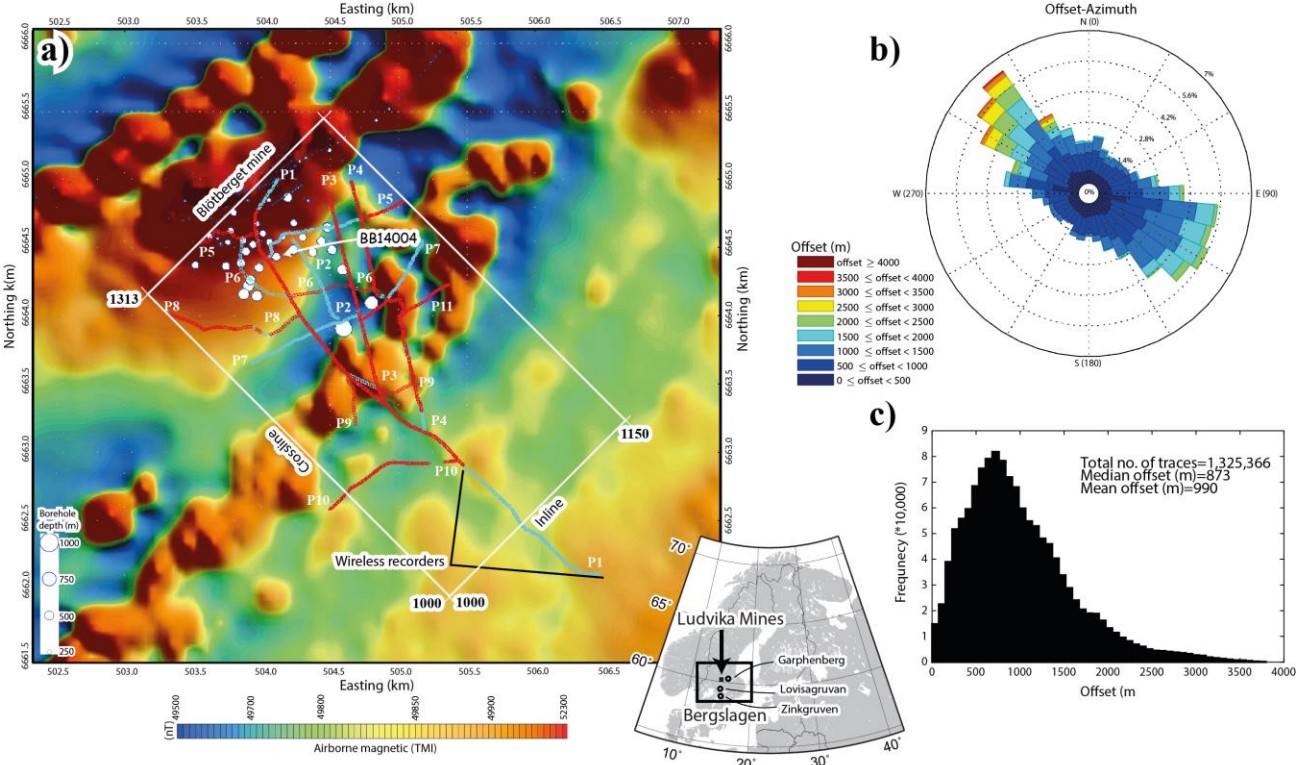


**Figure 1.** (a) Total-field aeromagnetic map of the Blötberget mine in central Sweden, and the sparse seismic survey area. P1-11 were setup as part of the survey (fixed geometry). The red dots show the shot locations (1056 points) and in blue are the receiver locations (1266 points). P10 and P11 were added later to improve fold and azimuth coverage and comprise only shots. Along P1 several 2D profiles have since 2015 been acquired (landstreamer survey in 2015 and conventional plant-type geophones in 2016). Downhole logging data from BB14004 is

presented in this study. (b) Offset-azimuth coverage showing a narrow azimuth survey setup for primarily targeting the known southeast dipping of the iron-oxide deposits and (c) offset distribution showing a median offset of approximately 900 m for the whole dataset. Magnetic data were provided by the Geological Survey of Sweden.

According to Nordic Iron Ore, mineral resources at Blötberget are known striking in NE-SW for several hundreds of meters

and down up to 800 m (based on two deep holes drilled in the early 70s; see Fig. 1) in sheets of 10-50 m thicknesses. Estimated



tonnage is 45,4 Mt of 41,7% iron classified as measured, and 9,6 Mt of 36,2% iron as indicated. In terms of inferred resources, one can add another 11,8 Mt of 36,2% iron. The site may have much more potential, given that the lateral extent of the deposits is less known and there are currently no boreholes available deeper than 800 m, on both eastern and western parts of the survey area. These areas were obviously one of the main targets of the 3D seismic survey to provide insight if drilling these places would add up to the existing resources.

In terms of structures, the deposits dip moderately (40-50 degrees) towards the SE in repeated horizons seemingly concordant in the stratigraphy; at a depth of approximately 500 m they dip much more gently in a listric-form manner. The Geological Survey of Sweden has mapped a number of topographic and magnetic lineaments in the area striking dominantly in NNW-SSE although their nature and 3D geometry are uncertain. Recent hydrological tests in preparation for the feasibility of mining in the existing boreholes suggest a potential for similar trending structures (e.g., fracture systems) immediately south the Blötberget. However, not much depth constraints are available nor evidence of any clear offset (faults). Historical mine plans suggest also a NNW-SSE trending fracture system intersected at depth during the mining activities.

With the exception of a speculative antiform axis mapped nearly 20 km south of the Blötberget, no clear evidence of folding is present on public geological maps. Outcrops are scarce in the area making detailed mapping extremely difficult. In Blötberget, this is even more problematic since it occurs in a rather swampy lowland area (Blöt in Swedish means wet, Blötberget means wetland) with only a few locations south of the study area where few patchy outcrops are present. Knowing any structures and their geometries at depth are important, not only for future mining operations but also to help understand the geological settings at which the deposits are emplaced thus optimizing future drilling programs in the area.

## 3 Physical properties of iron-oxide deposits

A good understanding of reflection seismic response is better possible when physical properties are studied in-situ using downhole logging methods (Salisbury et al., 2000). Through years 2015-2016, six boreholes (400-550 m deep) were downhole logged using various probes; relevant for this study is the full-waveform triple sonic (Maries et al., 2017). Density measurements could only be done on core samples at 1 m interval. These studies showed (Fig. 2) that at the presence of suitable geometries and sufficient signal-to-noise ratio, direct targeting of iron-oxide deposits (magnetite and hematite) is possible and should be extremely helpful for deep exploration at the site. In particular, it was found that iron-oxide deposits due to primarily their large density contrast with the host rocks should allow strong seismic signal, a fact that was later also confirmed through 2D and cross-profile recording seismic studies (Malehmir et al., 2017a; Markovic et al., 2020; Maries et al., 2020).

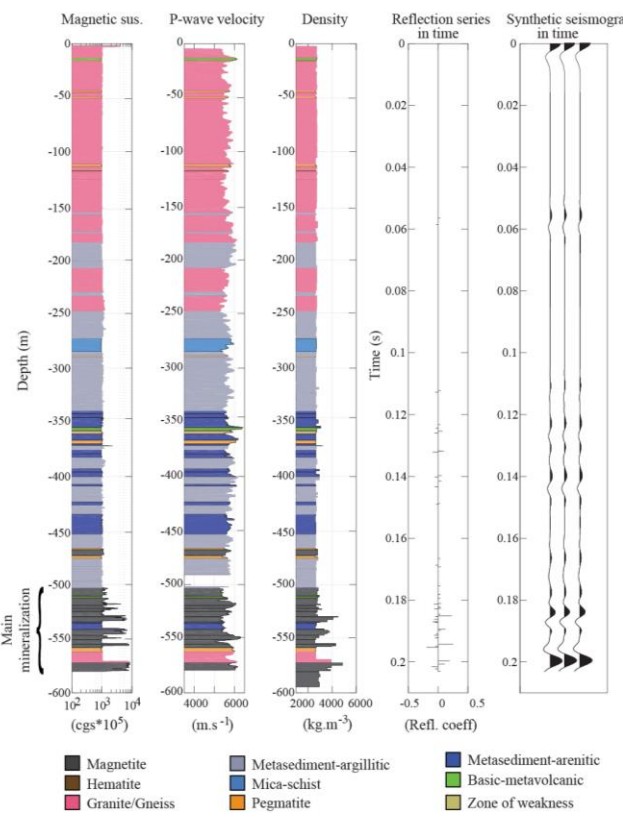

**Figure 2.** Example of downhole logging data from borehole BB14004 (Fig. 1) showing why a strong seismic response from the iron-oxide mineralization is expected based on the synthetic seismograms (70 Hz Ricker wavelet) generated from the data. Adapted from Maries et al. (2017).

## 4 Previous seismic surveys (2015 and 2016)

Before the acquisition of the 3D dataset, a number of 2D surveys were conducted at the site. Starting in 2015, a newly developed MEMs- (micro-electromechanical system) based seismic landstreamer was tested for a pilot deep-targeting work at the Blötberget site. Data were acquired using 100-MEMs sensors placed 2-4 m apart on the streamer (240 m long) and 75 wireless recorders placed north and south of the profile (fixed position, moved from the south to the north once the streamer progressed half-way length of the profile). A 500-kg Bobcat-mounted drophammer was used as the seismic source. In total, the streamer moved 9 times providing together with the wirelesses a nominal fold of 40. During 4 days, 3.5 km of seismic data along profile 1 (P1) were acquired using 1049 receiver and 533 shot locations. This combination allowed imaging the iron-oxide deposits down to 800 m depth (Malehmir et al., 2017a), a landmark for the use of landstreamers for such a purpose but also the potential of the seismic methods for their depth targeting.

In 2016, a more commercial-type survey using cabled-plant-geophones and the same drophammer seismic source but much higher fold (fold of 208 using 5 m shot and receiver spacing) was conducted (Bräunig et al., 2020; Markovic et al., 2020). A





cross-profile recording was also attempted using a fixed geometry along P1 (451 receivers) and a shorter perpendicular profile
(75 receivers spaced at every 10 m) along P6. Shots were recorded onto both profiles simultaneously. The 2016 experiment
showed deeper imaging of the iron-oxide deposits down to 1200 m. The cross-profile study (Maries et al., 2020) also suggested
a 300 m depth extension of the deposits as the ore body block models from borehole data. Potential geological structures were
also identified, but a 3D seismic survey was needed to account for the 3D geology of the site and its complex tectonic history.

Our earlier studies of 2D versus 3D seismic surveys (Malehmir et al., 2017b) were a further motivation not to push the 2D
interpretations far until a 3D seismic dataset becomes available.

## 5 Sparse 3D survey (2019)

As part of a large research-innovation project (Malehmir et al., 2019), a 3D seismic dataset (approximately 3 by 2 km) was
acquired during April-May 2019 using a fixed geometry comprising 1266 receivers (9 receiver lines, P1-P9) and 1056 shots

(10 shot lines, no shots on P2), and the 32t vibrator of TU Bergakademie Freiberg generating three shot records per location
with a 20 s long sweep ranging from 10-160 Hz (Figs. 1 and 3). A combination of 10 and 20 m receiver spacing was used
depending on profile location and after a pre-study of the fold and offset-azimuthal coverage using the equipment available to
this study. In the survey both cabled (P1 and P3) and wireless recorders (P1-P2, P4-P9) were used from two suppliers, Sercel
and Wireless Seismic (Fig. 4). The seismic profiles were planned to provide the best illumination angle, orthogonal to the

known strike of the mineralization. Table 1 details the main acquisition parameters of the 3D survey.

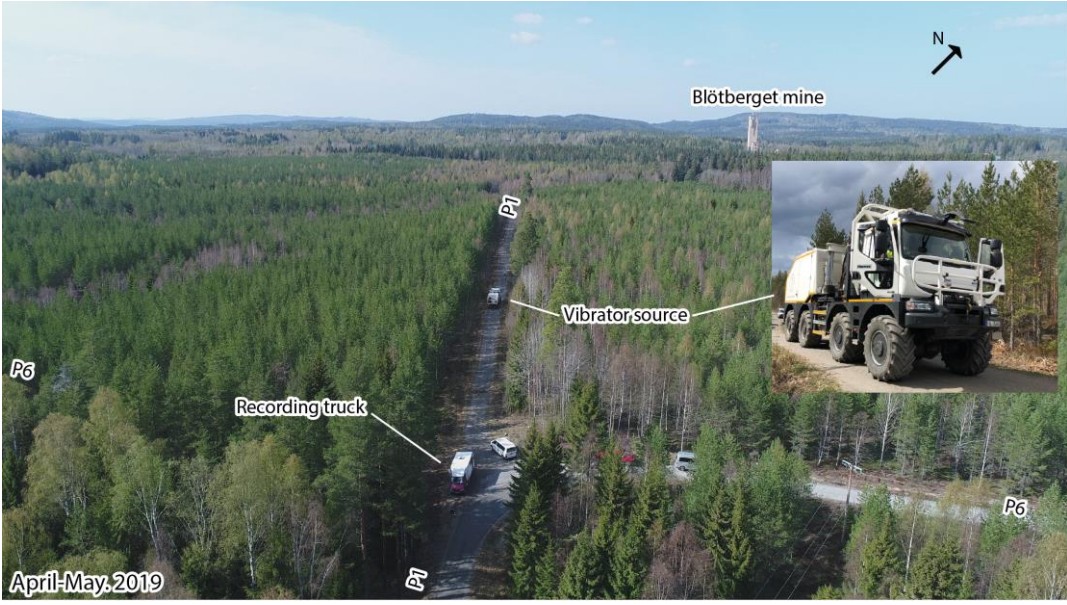

**Figure 3.** Field photos from the 3D seismic survey (April-May 2019) in the Blötberget mine. The survey was conducted using the 32t vibrator of TU Bergakademie Freiberg, 1266 receivers of which 414 were cabled (part of P1 and entire P3). The cabled profiles were used for live data quality control and sweep parameter tests. Earlier reported 2D profiles (years 2015 and 2016) were also along P1. Photos by
Alireza Malehmir.



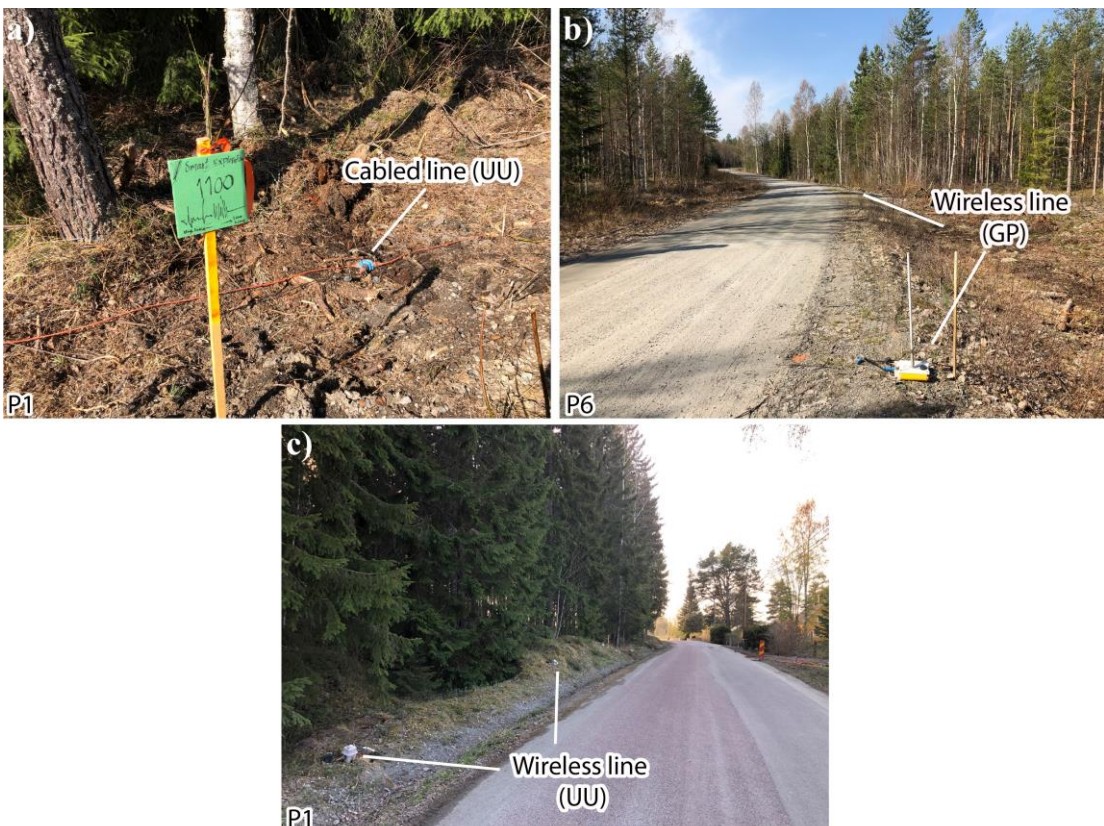

**Figure 4.** Field photos showing different recorder used during the 3D survey. (a,c) With the exception of P1 and P3 where a Sercel recording system was used (cabled and wireless along P1 and only cabled along P3), (b) all other profiles were acquired using RT2 Wireless Seismic requiring a maximum distance of 50 m for data crosstalk and live data harvesting. The choice of the recorders was solely dictated by their availability to the research team. UU: Uppsala University; GP: Geopartner. Photos by Alireza Malehmir.

## 5.1 Planned versus executed survey

Prior to the main survey, a number of visits had to be done in order to check for the location of shot and receiver lines. Earlier surveys in 2015 and 2016 were helpful in deciding which one of the profiles need to be acquired but also the geological questions concerning both depth and lateral extent of the deposits. We counted on 1500 receivers with a spacing of 10 m for both shots and receivers. However, this turned out to be impractical as nearly 250 extra receivers were needed during the survey. The planned survey counted for shots on the southern part of the area, which were not permitted at the end as well as shots along P7. Figure 5a,b shows planned receiver and shot profiles and CDP bins for 10 m and 20 m, respectively. The planning aimed to acquire a uniform source-receiver offset-azimuth coverage (Fig. 5d). However, in practice this was not possible and compromises had to be done. Figure 5c shows the executed survey and its fold coverage using a 10 m CDP bin size. Adding shot lines P10 and P11 as well as more shots along P8 helped to obtain a more azimuthally favourable 3D dataset with respect to the dip and strike of the known mineralization. With the exception of P1, P3 and P4, receivers were placed on



average every 20 m but shots where possible generated at every 10 m spacing. This setup helped to extend the survey area
towards the west compared to the planned one. The receiver setup had also taken into consideration a maximum distance of
50 m for the Wireless Seismic recorders hence the way the receiver profiles are connected in the 3D setup.

**Figure 5.** Planned versus executed 3D survey at Blötberget. Planned survey and fold coverage using (a) 10 m and (b) 20 m regular CDP
bins. (c) Executed survey and fold coverage using 10 m regular bins as used for the processing of the dataset. (d) Planned offset-azimuth
coverage. The executed survey is shown in Figure 1b. While the planned survey had more uniform offset-azimuth coverage, due to permitting
issues shots on the southern portions of P1, along P7 and parts of P5 could not be done. The planned survey also aimed for 1500 receivers,
which at the end only 1266 became available. The executed survey ended to cover more areas west of the survey area, which was important
in imaging westward extension of the deposits as discussed in the article.





## 5.2 Sweep parameter tests

We spent one day to test optimum sweep parameters given the swampy condition of the Blötberget area. The cabled P1 and
P3 lines were used for this purpose, but also for live data quality control during the main survey. About 70 different sweeps
were tested with different frequency ranges, drive forces and sweep lengths. Figure 6 shows a selection of shot gathers along
P1 (after cross-correlation) for various sweep ranges. A drive force of 60% was chosen to guarantee sufficient signal strength
for the envisaged target depth (<2000 m). The lower end of the tested frequency bands (10 Hz) was mainly dictated by the
vibrator specifications, while the upper end of the tested frequency band as well as the sweep length were more carefully tested.
Our analysis of the test shots convinced us to choose a linear sweep ranging from 10-160 Hz with a sweep length of 20 s to
allow more low frequencies to be generated compared to 17 s long sweeps. The choice of sweep length though has some
influence in the bandwidth of the signal with the 20 s appearing much flatter than 17 s a further reason why 20 s sweep length
was chosen. As for the cross-correlation, we tested with both, the pilot and the theoretical sweep, and with no question the
theoretical sweep produced the most convincing results hence used in the study and for the data quality control during the
survey and for processing the dataset.

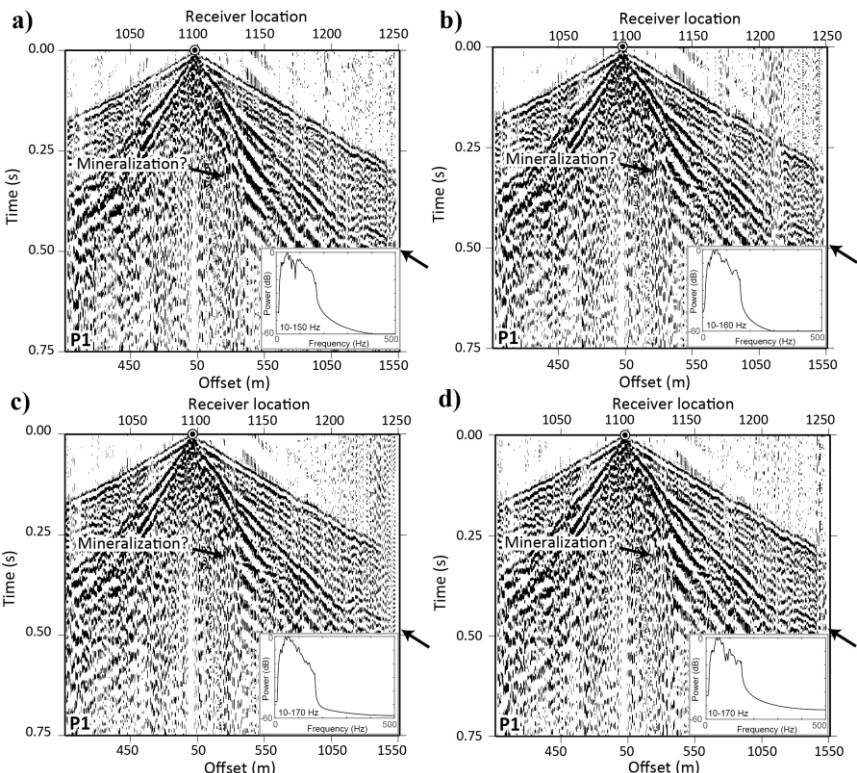

**Figure 6.** Shot gathers (after cross-correlation) and their corresponding amplitude spectra along cabled part of P1 showing sweep parameter tests (a) 10-150 Hz, 60% force and 17 s sweep length, (b) 10-160 Hz, 60% force and 17 s sweep length, (c) 10-170 Hz, 60% force and 20 s sweep length, and (d) 10-170 Hz, 70% force and 20 s sweep length. Both higher end frequencies of 160 and 170 Hz were judged suitable
however we decided to choose 160 Hz because it produced less ground-roll energy compared to 170 Hz. The arrow shows expected reflection from the mineralization also judged for choosing the sweep parameters.



## 5.3 Tailored processing workflow

Although the Blötberget is known for its swampy areas, human-made built-up places and several water streams can be found
in the area. The 3D seismic data show remarkable quality with nearly first breaks clear in all shot gathers. To show case this,
we present a shot gather after cross-correlation and vertical staking of three repeated shot records in Figure 7a. While near
surface conditions extremely vary from one profile to another, it is already possible to observe a strong reflection in nearly all
the receiver lines. This excellent data quality can also be judged as an argument on the choice of the source and sweep
parameters used in the survey. To illustrate that quality data were also acquired on other profiles, in Figure 8a we present a
receiver gather from P2 that was positioned along a major canal (most likely related to the past mining activities in Blötberget)
in the central part of the study area. Note that along P2 and most of P5 and P7 only receivers were possible to be placed hence
there are no shots for example from P2. Judging again from the quality of the first breaks and a strong reflection observed on
all the shot lines, one can already argue for the quality of the dataset hence a quality control for reliable interpretation of the
reflections in the final seismic volume. These record examples, however, show the challenges encountered with the processing
of such a dataset namely (1) strong presence of near-surface heterogeneity and (2) strong surface-wave noise. Therefore, data
processing had to be carefully tailored to handle these two issues through a carefully designed workflow and parameter
selections. Table 2 details the processing workflow and parameters applied to the 3D dataset. Here, we detail a few of the steps
given their significance and the sparse nature of the 3D dataset.

To correct for the effect of near-surface statics due to uneven overburden cover and velocity, approximately 1,2 million first
breaks were picked automatically and corrected manually where needed. 3D refraction static corrections using a two-layer
generalised reciprocal method (GRM) were first estimated with an RMS (root-mean-square) value of 5.5 ms. This was obtained
using a moderate smoothing parameter and using only offsets between 10-750 m. Three different rounds of iterations using
300, 200 and 100 m cells split into 4 triangles were consecutively used to obtain this solution and helped to narrow down the
RMS from 15 to 5.5 ms. While the 3D static corrections partly improved the coherency of the reflections (Figs. 7b and 8b) as
well as the first breaks, along some of the profiles such as P4 and P5 the resulting corrections were found too smooth to be
effective. This is clear if one carefully looks at the coherency of the first arrivals. Given the sparse nature of the 3D data, we
decided to also check if 2D refraction static solutions individually, estimated only for shots and receivers along a specific
profile could produce more coherent reflections than the 3D static solution. This was, in fact, the case especially along P4 and
P5 where even an RMS of 3 ms was obtained and the coherency of the first breaks is much better after the 2D static correction.
We decided at this stage to produce a number of brute stacks checking which of the solutions would better suit the dataset. The
best brute stack volume was obtained using a combination of 3D static corrections along those profiles where only receivers
or shots were placed (i.e., P2, P7, P10 and P11) and 2D static corrections for the remaining profiles (Figs. 7c and 8c).

To attenuate the strong surface-waves in the data, three different sets of filters were designed. First a broad bandpass filter was
applied to the data keeping frequencies between 30-150 Hz. This partly removed some of the surface-waves but a strong
portion still remained. To make sure surface-waves would stack out during the stacking step, a spectral equalization filter was





applied between 40-140 Hz. At a later stage an FK-filter had to also be applied without which surface-waves would have been still dominant in the final unmigrated volume. A top-mute function then was designed using the picked first breaks to make sure first arrivals would not leak as steep events in the final volume. Figures 7d and 8d show the effect of these processing steps on the shot gather example of P8 and the receiver gather example of P2. It is clear that the observed reflections are dominantly improved though there is still some surface-waves remained in the data.

For the CDP binning of the data, after a number of tests, a bin size of 10 by 10 m was chosen with inlines following the main direction of the receiver lines (intentionally positioned to favour the down-dip direction of the mineralization) to the NW-SE (Fig. 5c). According to Yilmaz (2001, Eq.1) using such a bin size, the maximum non-aliased frequency, $f_{max}$, for a reflector dipping at 60 degree, $a$, and using a mean velocity of, $v_{rms}$, 6000 ms$^{-1}$,

$$Dx \pounds \frac{v_{rms}}{4 f_{max} \sin a} \tag{1}$$

would be 170 Hz, which is comparable with the sweep frequencies used in the survey (10-160 Hz) and the maximum frequencies (140 Hz) kept in the data. A 20 m bin size would have implied 85 Hz maximum no-aliased frequency and considered too low for a hardrock setting.

We decided to employ a conventional NMO-based processing algorithm to the data to make sure a first-hand result is obtained and if any processing treatment is needed can be applied poststack or through an iterative process to the prestack

data. Velocity analysis was done although we found the range was not varying beyond 5800-6100 ms$^{-1}$. DMO corrections were attempted however due to the sparse nature of the data and offsets, it was already clear this would not be a suitable choice thus to avoid DMO artefacts, this process was excluded. NMO corrected gathers were then used iteratively to obtain surface-consistent reflection residual statics (two rounds) before stacking the data. The choice of the stretch mute was important as the main reflections were observed at wide-angle, implying a risk of loss if too low stretch mute was used. After

a series of tests and making sure that the first breaks were mostly muted, we chose a stretch mute of 50% and a diversity-type stack than just summing the traces (normal stack).

After generating the unmigrated stacked volume using CDP bins of 10 by 10 m, at inlines with low folds (i.e., margins of the survey area), notable surface-waves were observed but having horizontal characters in the inlines. To be able to attenuate them, data resorted to crosslines and these noises were selectively FK-filtered. For the migration, a phase-shift algorithm

worked best. Data were then time-to-depth converted using a smooth 1D velocity starting from 5900 ms$^{-1}$ in the top and 6100 ms$^{-1}$ at 1 second. Both unmigrated and migrated stacks were exported for 3D visualization and interpretations, and for checking if any time shift was needed to tie the reflections to known features observed in the existing boreholes (primarily iron-oxide horizons).



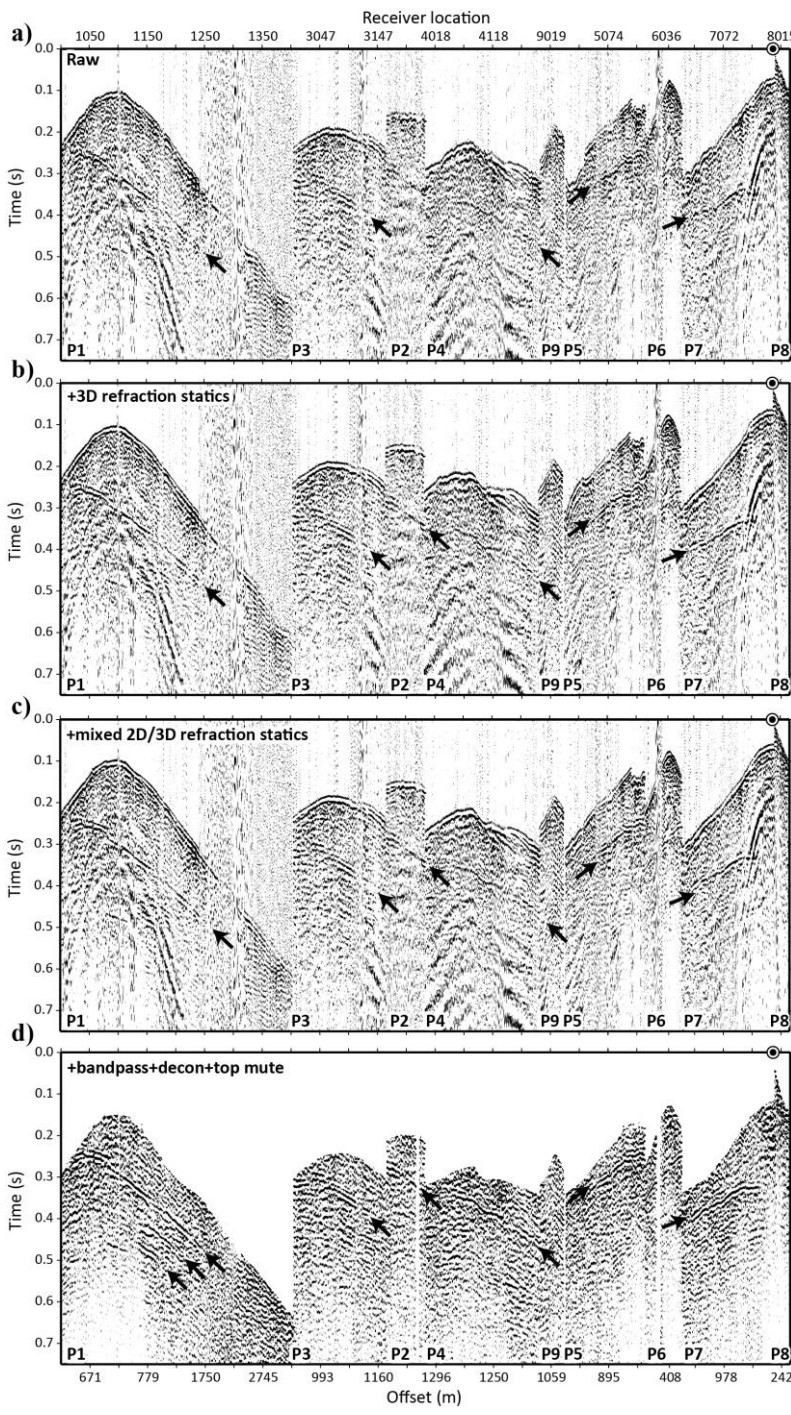

**Figure 7.** (a) An example of raw shot gather along P8 after (b) 3D static corrections, (c) mixed 2D and 3D static corrections (used for processing) and (d) bandpass filter, surface-consistent deconvolution, and top-mute. Note the increase in the signal quality in and especially coherency of the reflections marked using the arrows (we expect these to be from the mineralization from the earlier 2D surveys and downhole logging data).





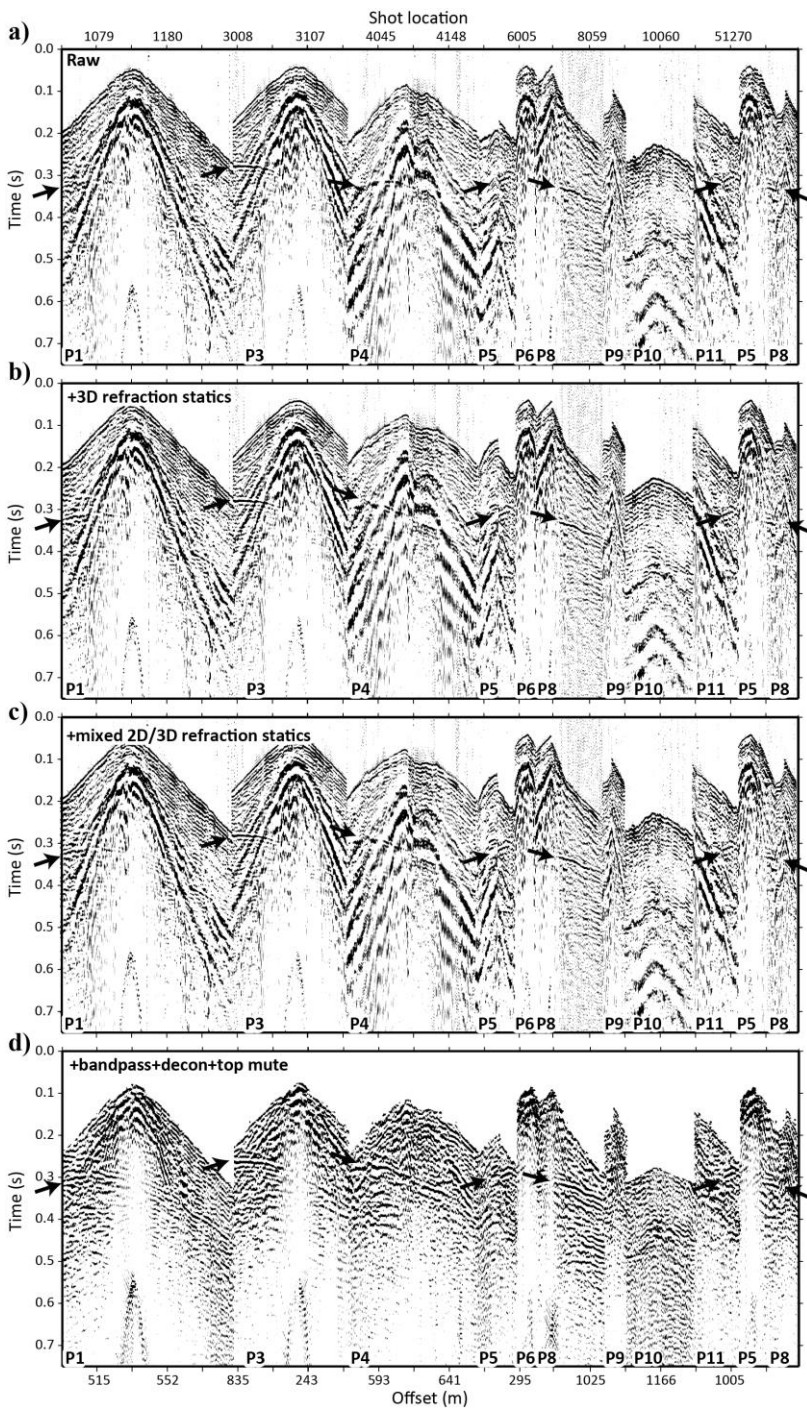

**Figure 8.** (a) An example of raw receiver gather along P2 after (b) 3D static corrections, (c) mixed 2D and 3D static corrections (used for processing) and (d) bandpass filter, surface-consistent deconvolution, and top-mute. Note the increase in the signal quality and especially coherency of the reflections marked using the arrows (we expect these to be from the mineralization observed on the earlier 2D surveys and downhole logging data). Note shots are only sorted based on their peg numbers as in between receivers were shots and named differently (multiplied by 10 e.g., 51270 is a shot along P5 and at the position between 5127 and 5128).





## 6 Results and Interpretations

### 6.1 Near-surface statics and historical tailings

Although the 3D static solution was not fully applied for the reflection data processing, given its potential value in understanding near-surface conditions (Malehmir et al., 2018) and potential geological features near the surface (e.g., fault and fracture systems) we present the results visualized in 3D together with the LiDAR (light detection and ranging) elevation map of the study area (Fig. 9). Figure 9a shows the high-resolution elevation map (1 m horizontal resolution and a few centimetres vertical resolution when compared with the DGPS of the receivers surveyed during the acquisition) of the study area. As evident, on the eastern part of the study area, immediately northern and eastern part of P4 and P5 an unusual built-up area occurs. We only realized this after the survey was completed although were warned about vibrating on the eastern portion of P5 prior to the survey due to the possible loose ground conditions. This built-up area is related to a historical tailing position (dry tailing) remained from the past mining activities. The near-surface velocities estimated from the 3D static solution (Fig. 9b) clearly depict the position of this tailing and two cascades of ponds south of it on the eastern portion of P4. P-wave velocities estimated for the overburden are on average around 700-1000 ms$^{-1}$, which is remarkably much lower than that of observed for glacial tills (usually around 1500-2000 ms$^{-1}$). Estimated bedrock velocity and depth (Fig. 9c, d) also provide linear features particularly one immediately north of a railroad where no more shots could be generated due to the permitting issues. This bedrock level lineament is also associated with a LiDAR topographic lineament (Fig. 9d) and most likely represent a major fault holding down much of the survey area and responsible for the swampy environment of the Blötberget.

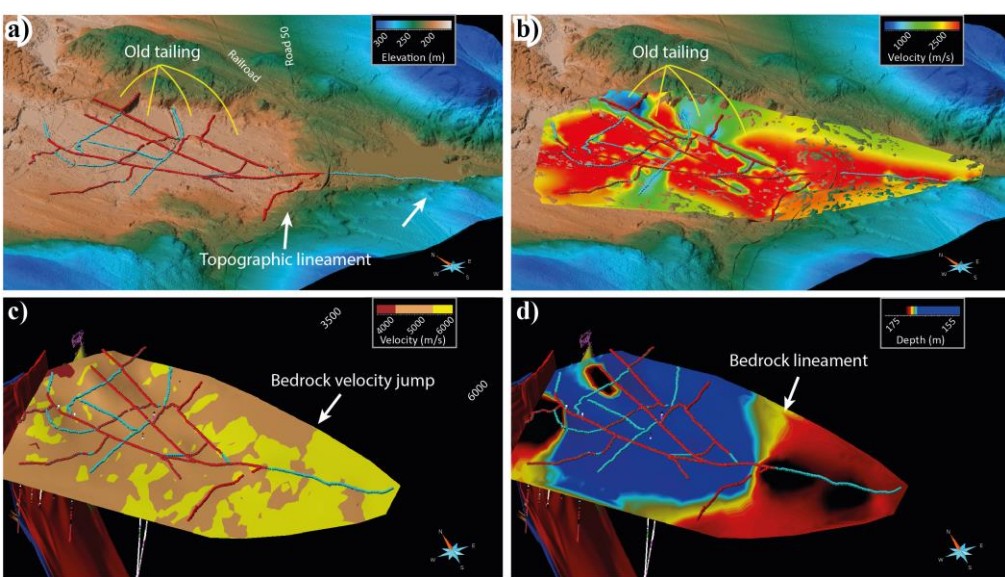

**Figure 9.** (a) LiDAR elevation map of a portion of the 3D survey area showing built-up roads and a historical tailing where P4 and P5 were positioned. (b) Near-surface velocities derived from the 3D refraction static solution showing the clear position of the tailing as low-velocity materials and the two separate ponds immediately south of it. (c) Bedrock velocities and a sharp jump in the velocity south of the survey area and (d) bedrock level showing a clear lineament (higher ground) immediately south of the road 50 (and the railroad). We attribute the bedrock lineament to a (or set of) normal fault.



### 6.2 Unmigrated stacked volume

To provide a measure for the quality of the migrated volume, it is important that unmigrated stacked volume is assessed and
any migration artefacts are recognized, or features, to be interpreted and trusted. For this purpose, we present a series of slices
through the unmigrated volume in Figure 10. The slices show a timeslice (320 ms), an inline (1065), and a crossline (1221)
from the unmigrated stacked volume. The inline section shows a series of southeast-dipping reflections down to 400 ms (M1
and M2) after which the quality of the reflections or their continuity is weak. The reflections appear to end with two sets of
diffractions (D1 and D2) and crosscut by a moderately northwest dipping reflection (F1). In the corresponding crossline
section, these reflections appear to be two sets having a gentle dip towards the west with a middle steeper reflection (F2)
crosscutting them. A careful analysis of these features in conjunction with the timeslice suggests that the two sets of reflections
have different strikes and dips. These reflections, given their strong amplitude are likely from the iron-oxide mineralization.
The diffractions imaged nearly at the tails of the reflections at approximately where the crosscutting northwest-dipping
reflection intersects them may suggest an abrupt termination of the mineralization or a sudden change in their geometries. We
discuss this later where implications for deep targeting is presented in the study area.

### 6.3 3D image of the mineralization and potential resources

Figure 11 shows a series of slices extracted from the migrated and time-to-depth converted stacked volume. The slices show
two clear sets of reflections (M1 and M2) extending down to approximately 1200 m depth with a lateral extent of approximately
500 m each. In order to better interpret these features, we visualised the slices in 3D with the known deposit block models as
shown in Figure 12. The 3D visualization helps to associate the strong southeast-dipping reflections to the known
mineralization providing a better estimate of how much more vertically and laterally they may extend beyond what has been
modelled by boreholes. To avoid a manual interpretation of the reflections, through a thresholding exercise and adjusting
amplitudes that match better the intersection of the mineralization in the boreholes, we extracted regions of high amplitudes
associated with these reflections (Fig. 13a). Before doing this, all amplitudes were squared to account for both positive and
negative (peak and trough) values. This helped to identify automatically a minimum 300 m vertical and lateral extent than
what was provided by the borehole data (Fig. 13b). In particular, the possible lateral extent of the mineralization towards the
west and a complicated reflectivity with curved (F2) and submerging features are great information from this 3D visualization
of the results.



**Figure 10.** A series of slices (a) timeslice, (b) inline and (c) crossline through the unmigrated stacked volume showing two major sets of southeast-dipping reflections (M1 and M2) terminated by a northwest-dipping one (F1) at where two sets of diffractions (D1 and D2) are present. The crossline section shows these two sets of reflections have gentle but different dips towards the west. A reflection (F2) appears to cut through these two sets.






**Figure 11.** A series of slices (a) depth slice, (b) inline and (c) crossline through the migrated and time-to-depth converted stacked volume showing two major sets of southeast-dipping reflections (M1 and M2) terminated by a northwest-dipping one (F1). The crossline section shows these two sets of reflections have gentle but different dips towards the west. F2 reflection appears to submerge (downlap) with M1. In the footwall of M1 and M2, another submerging reflection (M3) appears although weaker.




## 6.4 Structures and their implications

At least four sets of reflections apart from the main southeast-dipping ones could be identified. The first and most notable one is the F1 northwest-dipping reflection appearing usually weak in the volume and only traceable in a few inlines (Fig. 12a). This reflection was picked from different inlines where visible. A surface was then generated from the picks (Fig. 12b).

Because there are no shots on the southern part of the 3D volume, the reflection could not be imaged all the way to the surface and if it does. Therefore, a plane was fitted to the extracted surface in order to better find where it would project to the surface and if there is any corresponding feature associated with the reflection. This reflection was also observed in the earlier 2D surveys (Maries et al., 2020; Markovic et al., 2020) as it will be compared later. However, the 3D seismic data provide better information about its 3D geometry. The reflection strikes approximately N-6°-E and dips around 25 degrees towards

dominantly the west (Fig. 12d). Similar trending features have also been reported in the mine during the time of mining activities but rather to mainly fracture systems (no clear offset reported). We however think from its intersection with the mineralization at approximately 1200 m depth, this feature is likely a major fault system. While there is no marker horizon providing information about its nature, we think it is mainly a normal fault (multiphase but down-faulting still dominant). Multiphase faults are abundant in the Bergslagen area (Stephens et al., 2009; Malehmir et al., 2011) due to its complex tectonic

history. An alternative, but weakly supported interpretation for F1 reflection, could be a continuation of the mineralization forming a large curved synform-shaped system. To provide a definite answer on the origin of this reflection, boreholes are needed, and these are not currently available on the southern part of the study area nor we can connect this to a certain feature or outcrop. The best surficial evidence is a valley-type feature observed in the LiDAR data along the strike of the modelled F1 (Fig. 12d).

A second northwest-dipping reflection, albeit much weaker and slightly steeper, can also be identified disturbing partly the continuity of the main southeast dipping reflections (Fig. 12a). We were able to pick this reflection by looking into only the phase of the data and extract a new surface from the picks in different inlines (F3). Interestingly, F3 reflection (Fig. 12b) appears at where the refraction static solution suggests a bedrock lineament (Fig. 9d), but also the LiDAR data show an elevated ground surface (Fig. 9a). Much of the area north of this lineament is lowland and where the swamps or wetlands are present

(i.e., Blötberget). We, therefore, interpret F3 reflection, which strikes nearly N-S and dips approximately 45-50 degrees towards the west as a dominant normal fault responsible for much of the lowland in the Blötberget region.

We also identified two smaller reflections, one with similar dip and trend as the mineralization (L1) but likely from the contact of volcanic with the intrusive rocks, as it could be matched in boreholes (Fig. 12c,d) and another down-lapping with the interpreted reflections from the mineralization (F2). The F2 reflection (Fig. 12) may be an intrusion or form a folded structure

with the mineralization. Due to the lack of any borehole west of the survey area, we cannot conclusively present a geological interpretation for this feature, however, if it is from the mineralization, it may reach closer to the surface towards the west making the western part of the study area strongly prospective.





Similar to the earlier studies underlying the main mineralization, there are weak but parallel reflections that may present additional resources in the footwall of the known ones (e.g., M3 in Fig. 11). These reflections appear to merge with the main
reflections from the mineralization and are only 200-300 m deeper in their footwall. Therefore, any future deep drilling should continue for 200-300 m deeper to check if these reflections are associated with mineralization. Table 3 summarizes the main features of the 3D seismic volume in terms of geometry and our interpretation of their origins.

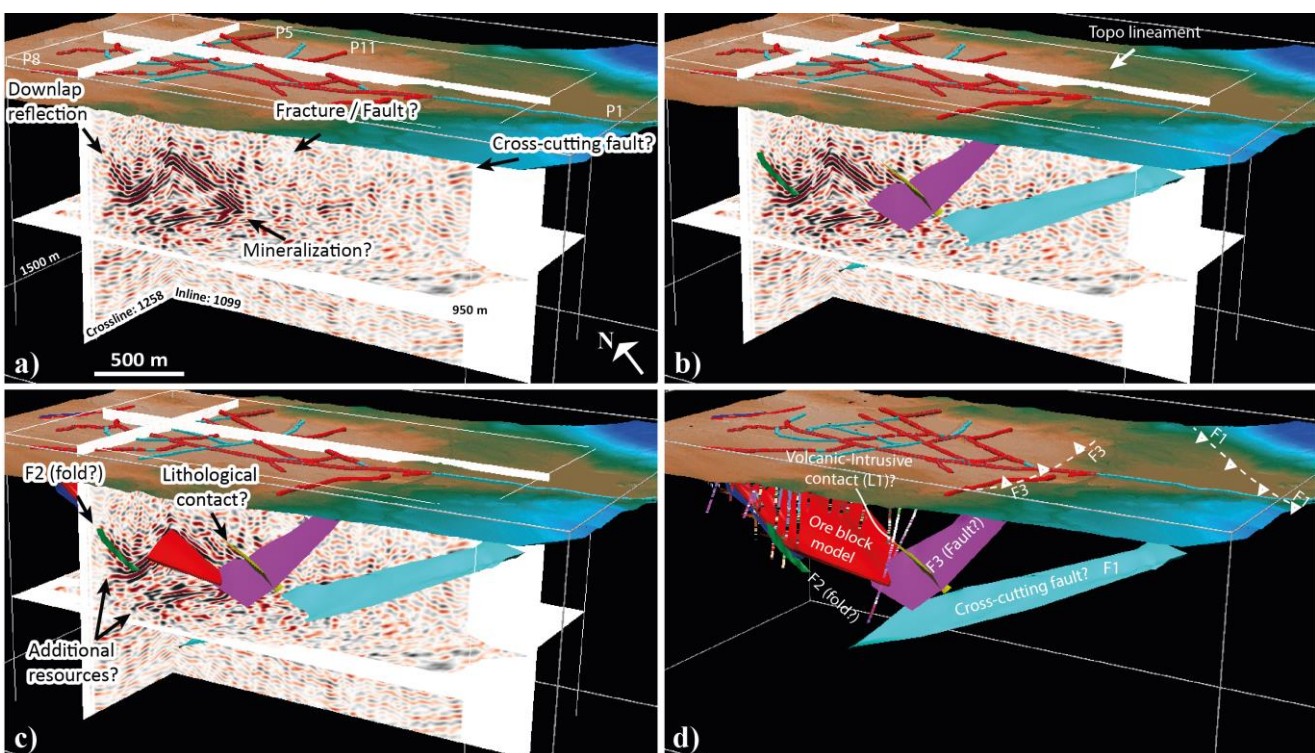

**Figure 12.** 3D views from the migrated and time-to-depth converted stacked volume showing (a,b) a series of southeast-dipping strong
reflections and a number of northwest dipping one and how (c,d) they have been picked to represent various surfaces within the volume. The red surface is ore block model derived from borehole data. F1-F3 are interpreted to be either from fault and fracture systems or folds. Note also how F1 and F3 may be correlated with the topographic lineaments observed on the LiDAR data. Reflection L1 is interpreted to be from the contact between volcanic and intrusive rocks.

## 7 Discussion

The Ludvika 3D sparse seismic survey only used existing roads and forest tracks for generating shots and proved to be instrumental in delineating 3D geometry of the iron-oxide deposits providing information on potential additional resources in the down-dip and laterally for a minimum of 300 m in each direction (Fig. 13). Assuming an average thickness of 30 m and a density of 3800 kgm$^{-3}$, this would add approximately 10 Mt potential additional resources that are worth to be drilled for mineral resource assessments. Tying the high-amplitude regions with the borehole data (Fig. 13a) was a great way of extracting
places where iron-oxide deposits could be present (Fig. 13b). The 3D seismic survey, to our best knowledge, is the first





published account of using 3D seismic methods in Sweden for deep targeting and mineral resource exploration. The earlier 2D surveys while provided also key information (Marries et al., 2020), the 3D geometry of the structures are much better defined in the 3D survey (Vestrum and Gittins, 2009; Malehmir et al., 2017b). For example, the F1 structure should be an out-of-the-plane feature, that intersected the earlier 2D profiles (Fig. 14a,d) obliquely with a strike similar to those reported in the

underground mine (when operational), hence important for future planning of the mine. It is still unclear if F1 is a fracture system or a fault, although our preferred interpretation is a major normal fault possibly contributing to the repeated reflectivity in the footwall of the known (drilled) deposits. Given the narrow azimuth nature of the 3D survey (dominantly NW-SE), structures like F1 would require better sampling in the E-W direction and this likely contributed into its weak imaging than reflecting its actual physical property contrast. Structures such as F3 and M-M3 were also imaged (Fig. 14b,c,d) in the earlier

2D profiles but their definition and submerging nature is better defined in the 3D volume. This illustrates why 3D seismic surveys are better suited for complex geological structures in hard rock settings.

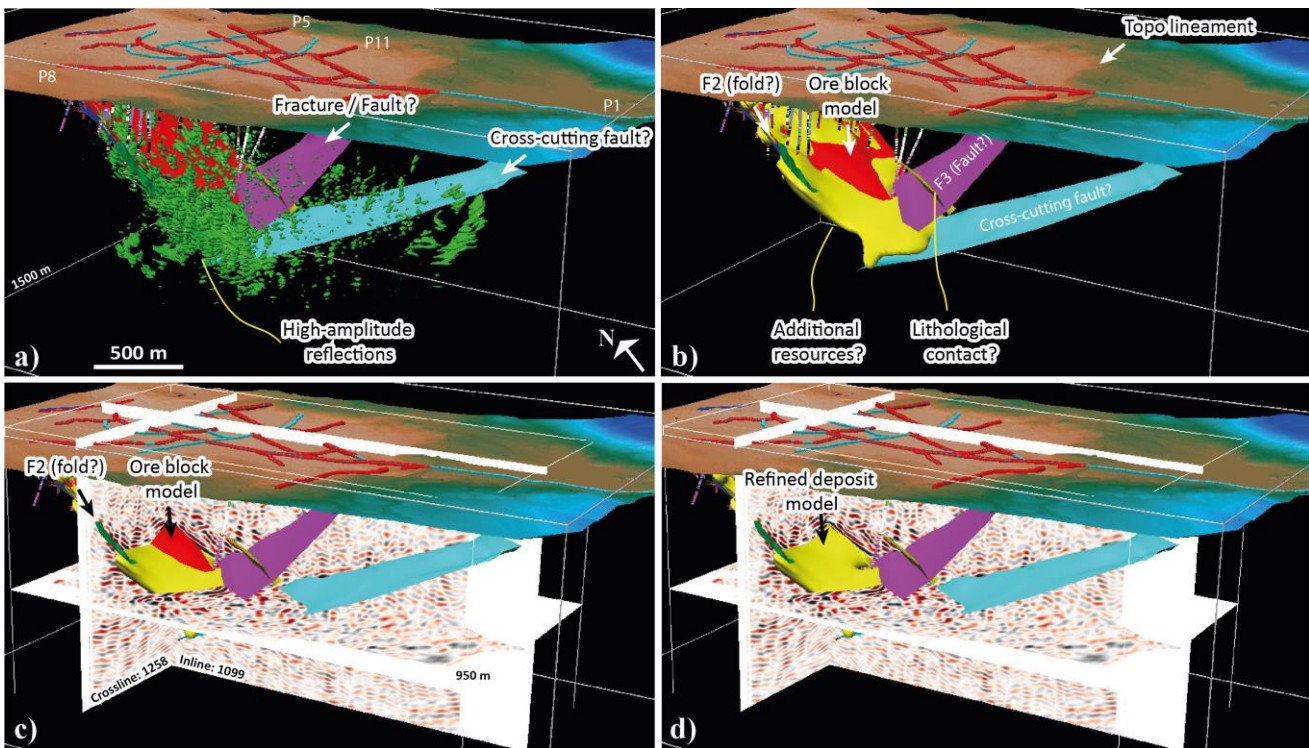

**Figure 13.** 3D views showing how additional potential resources were identified from the 3D seismic volume. (a) High-amplitude reflections were first through thresholding extracted as iso-shells (green shells) and then (b) matched with the boreholes intersecting the mineralization

and merged with the ore block model to produce an updated resource model (yellow shell). (c,d) Similar views as (a,b) showing how the refined resource model match the strong southeast-dipping reflections and how far to the west and vertically they may extent; our estimate is a minimum of 300 m.

In terms of data resolution, the seismic data contain a good band of useful frequency content with a dominant frequency of

about 70-75 Hz, which based on an average medium velocity of 6000 ms$^{-1}$ would be equivalent to approximately 80 m




wavelength. Given the two distinct diffractions (D1 and D2) observed in the data (Fig. 10), we estimate a vertical resolution on the order of 20 m implying that the strong southeast-dipping reflections are each likely from a body of mineralized horizon than their top and bottom. The diffractions were also studied for the velocity they would require to collapse to a point (using the diffraction hyperbolic equation), which was around 5900 ms$^{-1}$, consistent with the velocities picked for the NMO

corrections, migration and time-to-depth conversion. There are a few other notable diffractions in the unmigrated stacked volume that have potential to be further studied, to provide geological information and scale of heterogeneity in the dataset and we recommend this to be done in future studies.

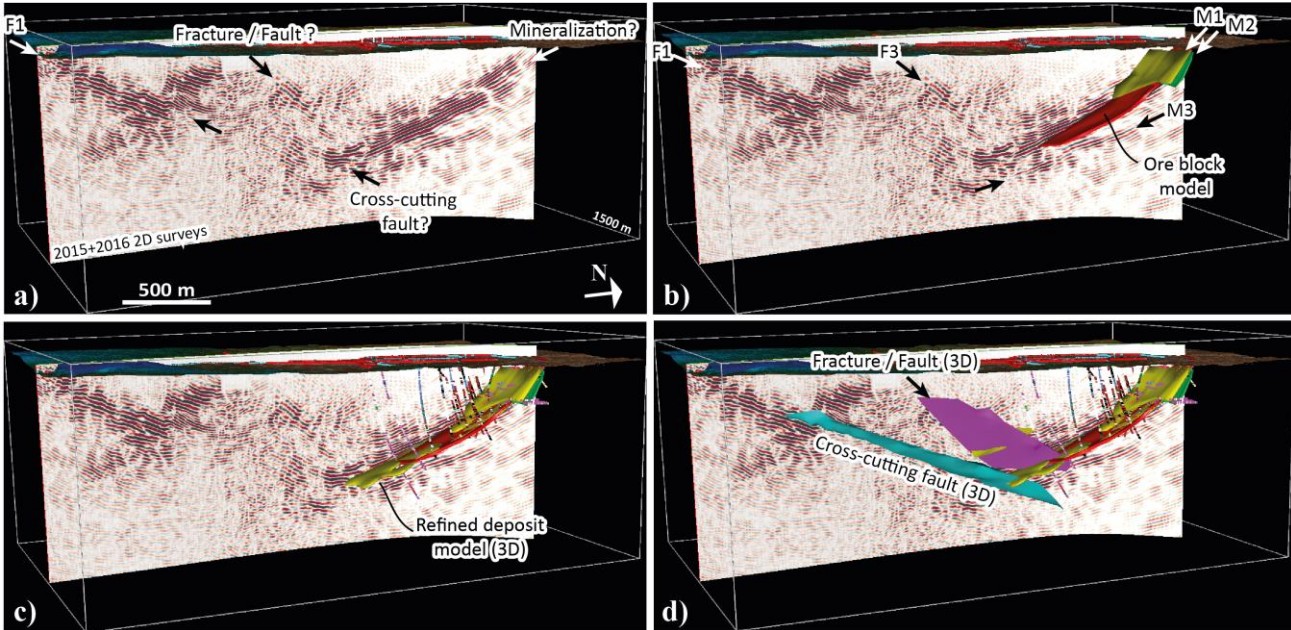

**Figure 14.** 3D views from (a) the earlier 2D surveys (Markovic et al., 2020) and (b-d) how various features extracted from the 3D volume
manifest themselves in the 2D data. In particular, note that the F1 and M3 are better defined in the 3D volume showing a westward plunge for F1 and a submerging character for M3 implying they have a major out-of-the-plane nature.

As for the data processing of such a dataset, DMO corrections were applied and partly helped to better image the F1 reflection in a few inlines but overall produced a noisy volume that required more processing treatments (due to partial migration

artefacts) than the volume without the DMO corrections. Hence DMO corrections were excluded from the processing workflow. This is not surprising, as the 3D dataset is sparse, narrow azimuth and contains irregular offset-azimuth traces (Beasley and Klotz, 1992; Ronen et al., 1995; Vermeer et al., 1995). We did not consider any trace interpolation given the complexity of the reflections and their crosscutting natures. While more receivers and shots could have solved this issue, we were limited to the number of recorders made available for the survey, and we did not want to make additional shot lines due

to cutting forests, environmental and ultimately cost issues. More shot points however on the southern part of the study area would have been useful in order to obtain a deeper image of the mineralized horizons (if so) and a better definition of F1 reflection near the surface. These shot points were initially planned however due to the permitting issues for vehicle heavier



than 12t, the seismic vibrator (32t) could not be used there. As well noted by Bouska (1997), 3D surveys should be designed for providing details that allow geological interpretations for economic success. If 3D seismic surveys are too pessimistically

designed using only the worst scenarios, and to account for too unknown complex geology, we might acquire much less 3D surveys and miss opportunities. In the case of Blötberget, the combination of 2D and downhole logging surveys, the use of borehole information and ore body block model helped to plan the 3D survey, that provided a great added-value with likely 40% of that being acquired full azimuth and high fold (plus 100) should have this especially been conducted by commercial contractors using rule-of-thumb approaches.

In terms of 3D acquisition footprints (Vermeer, 1998; Gulunay et al., 2006; Cheraghi et al., 2012), we only observed strong surface-wave noise remained at the margins of the volume due to having a lower fold. No other footprints were noted. It is however worth mentioning one reason to reduce this could be because of the use of the diversity stack for CDP stacking. The choice of a mix of 2D and 3D statics was important in order to provide an improved image of the reflections, and we suggest this to be considered for complex datasets as 3D statics may only provide a smooth and long-wavelength static solution

allowing a not necessarily higher resolution imaging. When we considered this, a careful analysis was done to make sure that the shifts from 2D to 3D are not so significant to make sure a mix then was plausible. Two runs of surface-consistent residual static corrections then helped to adjust any remaining mismatches. This approach was also attempted using the 3D static solution but was not as successful as when the mixed solutions were employed.

Parallel to this study, 3D focusing prestack depth imaging algorithms (Buske et al., 2009) are being applied to this data set but

the corresponding results are not the focus of this study. Instead, this paper provides an inventory publication on the nature of the 3D seismic survey, how it was implemented and how a tailored processing workflow provided good information for deep exploration and targeting in the site. What is worth emphasizing is that the 3D seismic survey also provided additional information on the location and characteristics of the historical tailings, bedrock lineaments that were complementary and further showcase why 3D reflection seismic data are an asset for mining and mineral exploration and why they should be tried

more often for both purposes (Malehmir et al., 2018).

## 8 Conclusions

A sparse 3D seismic dataset was acquired in the Blötberget mining area of central Sweden for deep targeting and better understanding geological structures hosting iron-oxide deposits at the site. The survey benefited from careful planning, downhole logging data, earlier 2D and cross-profile recording surveys as well as knowing roughly 3D geometry of the deposits

from borehole observations some of which were from the early 70s. The survey was a joint effort of a few organizations putting their 1266 seismic recorders and a 32t seismic vibrator together. While data quality is very good for being acquired in a hard rock setting, processing work was challenging due to the extreme ground conditions from built-up roads, historical tailings as



well as swampy parts, producing a significant amount of surface waves to handle in this narrow source-receiver offset-azimuth dataset.

In particular, the choice of refraction static corrections and attenuating surface-waves were the key parts of producing quality seismic volumes (both unmigrated and migrated stacks). The 3D seismic volume helped to image down-dip and lateral continuation of the mineralization for a minimum of 300 m especially on the western part of the survey area where there are currently no boreholes available. If this interpretation is correct and that the strong reflections originate from iron-oxide horizons, our estimate suggests potential additional resources of about 10 Mt at depth that can be accounted for or argued for

further drilling and resource assessments. The 3D seismic survey also provided oppositely dipping features, i.e. westerly dipping, interpreted to be from major normal faults responsible for much of the lowland of the Blötberget and the repeated reflectivity pattern observed in the footwall of the known deposits. These structural features as well as the potential resources should be the targets of future drilling to maximize the value the 3D seismic survey provided. As the by-product of the processing work, the 3D refraction static solution helped to map the historical tailings and areas immediately south of it

illustrating why 3D seismic surveys have much more value than providing targets for mineral exploration. Given the positive results of the survey, we encourage mining companies in Sweden to attempt such a survey, as the added value is reasonably high.

## 9 Acknowledgments

This study was conducted through the Smart Exploration project. Smart Exploration has received funding from the European

Union's Horizon 2020 research and innovation programme under grant agreement No. 775971. We thank Nordic Iron Ore and several young professionals from Uppsala University, Geopartner, TUBAF and TU Delft who worked and supported us during the survey to acquire this research-innovation dataset. In particular we thank Nordic Iron Ore for collaborating with us in the project and this study.

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

**Table 1: Main acquisition parameters of the Blötberget 3D seismic survey, April-May 2019.**

| | |
|---|---|
| **Survey parameters** | |
| Recording system | Sercel 408 (Uppsala University) & Wireless Seismic (Geopartner) |
| No. of receiver lines | 9 (P10 and P11 are only shot lines) |
| No. of shot lines | 10 (P2 is only a receiver line) |
| Receiver interval | 10-20 m (10 m along P1, P3 and P4) |
| Shot interval | 10 m |
| Maximum source-receiver offset | ~ 3700 m (along P1) |
| Survey area | 6 km$^2$ |
| Source | 32t vibrator (TU Bergakademie Freiberg) |
| CDP bin size | 10 m (inline) by 10 m (crossline) |
| | |
| **Spread parameters** | |
| Receiver spread array | 1266 live receivers, fixed geometry |
| Receivers | Single 7.5 cm spike 10 Hz (only at a few places along P1 4.5 Hz) |
| Recorders | FDU along P1+P3, RAU along P1, Wireless Seismic along all other profiles |
| Source sweeps | 10-160 Hz (20 s linear) |
| No. of sweeps per points | 3 sweeps/shot points |
| No. of shot points | 1056 |
| Sampling rate | 1 ms (Sercel 408) & 2 ms (Wireless Seismic) |
| Geodetic surveying | DGPS combined with national LiDAR |







**Table 2: Principal processing steps applied to the Blötberget 3D dataset (2020).**

| Step | Parameters |
|---|---|
| 1. | Read 30 s uncorrelated SEGD data and resample to 1 ms |
| 2. | Cross-correlate with the theoretical sweep (3 s output) |
| 3. | Vertical stacking of repeated shot records (1 s output for processing) |
| 4. | Extract and apply geometry (CDP bin size of 10 by 10 m after several tests) |
| 5. | Inspect data quality and inconsistency, correct for bad positions and elevations using LiDAR data |
| 6. | Trace editing |
| 7. | Pick first breaks: full offset range (approximately 1,2 million traces); automatic neural network algorithm but manually inspected and corrected |
| 8. | Refraction static corrections: (mix of 2D and 3D used); datum 210 m and replacement velocity of 5800 m.s$^{-1}$ |
| 9. | Bandpass filtering: 20−30−150−160 Hz |
| 10. | Spectral equalization: 20−40−130−150 Hz |
| 11. | FK-filter (only one dip) targeting strong surface-waves |
| 12. | Air-blast attenuation (330 m.s$^{-1}$) |
| 13. | Trace balance using data window |
| 14. | Top mute: 20 ms after the first arrivals |
| 15. | Velocity analysis (iterative): every 5th inlines |
| 16. | Residual static corrections (iterative): two rounds |
| 17. | Normal moveout corrections (NMO): 50% stretch mute |
| 18. | Stack (diversity) |
| 19. | $f_{xy}$-deconvolution |
| 20. | Bandpass filter: 10−30−110−140 Hz |
| 21. | FK-filter (only one dip) targeting remaining surface-waves in the crosslines |
| 22. | Migration: using 1D borehole velocities, 3D phase-shift |
| 23. | Time-to-depth conversion: 5900-6100 m.s$^{-1}$ |
| 24. | Export for 3D visualization (both unmigrated and migrated stacks) |

**Table 3: Identified main reflections and their 3D natures.**

| Reflection | Strike | Dip | Origin |
|---|---|---|---|
| **F1** | N-6°-E | 20-25°/W | Fault generating diffractions too (D1 and D2) |
| **F2** | N-2°-E | 45-50°/NE | Fold (submerging with mineralization, M1 and M2) |
| **F3** | N-2°-E | 45-50°/W | Fracture or normal fault system causing the lowland and swamps in the Blötberget |
| **M1** | N-25°-E | 25-30°/SE | Mineralization |
| **M2** | N-25°-E | 25-30°/SE | Mineralization |
| **M3** | N-25°-E | 20-25°/SE | Likely mineralized and merging with M1 and M2 at approximately 1200 m depth |
| **L1** | N-25°-E | 25-30°/SE | Lithological contact (volcanic-intrusive) |