# Peer review of "Sparse 3D reflection seismic survey for deep-targeting iron-oxide deposits and their host rocks, Ludvika Mines-Sweden"

_Solid Earth, 2020_

## Referee Comment (RC1) · Anonymous Referee #1 · 27 Sep 2020

The manuscript shows the 3D seismic data acquisition for mining exploration in Sweden. Although, this survey is not a conventional 3D dataset similar to those acquired in Canada which would provide the opportunity to apply more advanced processing steps (e.g., DMO or PSTM), the authors were able to address the narrow azimuth nature of this survey and showed reasonable results.

---

## Referee Comment (RC2) · Anonymous Referee #2 · 10 Dec 2020

This manuscript is well written, it can be accepted for publications only subject to minor revisions (1) The abstract should be prepared more succinctly, in particular, it is unnecessary to introduce the parameters for seismic data acquisition in so many details. (2) Line 25. What do you indicate for 10 Mt additional resources? Iron ? How did you estimate this potential source? Please indicate the grade of this ore deposit used in the estimate. (3) To be outspoken, I did not see significant improvements of 3D survey compared to 2D survey in Figure 14. Could you please explain this improvement better?

---

## Author Comment (AC2) · 10 Dec 2020

We appreciate the positive and favorable review and comments of the reviewer. They are all to the point and will be addressed in our revised manuscript. We here detail our response to the particular questions raised by the reviewer.

(1) The abstract should be prepared more succinctly, in particular, it is un- necessary to introduce the parameters for seismic data acquisition in so many details.

We will shorten the abstract and remove some of the acquisition details. There are also information about the assay and deposits that will be moved to the geology section.

[Figure]

(2) Line 25. What do you indicate for 10 Mt additional resources? Iron ? How did you estimate this potential source? Please indicate the grade of this ore deposit used in the estimate.

Sorry if this was unclear; the ca. 10 Mt estimated additional resources from the seismic data/volume, is based on a lateral and vertical extent of 300 m (300 m by 300 m) and a thickness fo 30 m. We have the lateral and depth extent controls from the seismic volume however the thickness is unresolved and is only based on an assumption that the sheet-like deposits have this thickness where drilled through at other locations. The density was assumed around 4000 kg.m3, a reasonable density for economic Fe-content rocks (40-50% Fe). Obviously, this is then is not Fe content rather tonnage of the rocks worth mining. We will detail this in the revised manuscript.

(3) To be outspoken, I did not see significant improvements of 3D sur- vey compared to 2D survey in Figure 14. Could you please explain this improvement better?

Well-noted and our bad! Figure 14 is only showing the earlier 2D section but with features/surfaces extracted from the 3D volume. Our intention was to show how the 3D cube helped to identify features that are not in the plane of the 2D section. We will rework the figure cation and text to clarify this. Sorry and thanks for noting this.

---

## Referee Comment (RC3) · Anonymous Referee #3 · 13 Jan 2021

This paper is reporting the results of a new (non-conventional) 3D seismic data acquisition, performed for mining exploration in Sweden. The authors well show how the combination of a careful and customized survey planning and optimized processing allow to achieve better results in comparison to a standard 2D survey, with reasonable costs. The manuscript represents a good contribute for the scientific progress in mining exploration, currently a very "hot" topic, presenting this case history as a reference example for extending this approach to other cases. This paper fits the scope of SE. It is complete, well-structured and well written, even if I've found a bit difficult to follow the interpretation details on the figures. Therefore, some updates on the figures are required. This manuscript can be accepted for publication after a minor

revision. Comments are attached in the pdf file.

Please also note the supplement to this comment:
https://se.copernicus.org/preprints/se-2020-141/se-2020-141-RC3-supplement.zip

—————————————————————

---

## Author Comment (AC3) · 15 Jan 2021

Response to the referee#3 comments: We thank the positive and favourable review and comments of the reviewer and that the topic is considered "hot" these days. We agree with the reviewer and this is why we wanted to showcase how-to and example works in this current manuscript. We do not detail every detailed point rather the most important comments and how we will address them in the revised manuscript.

General comments:

1) shorten the abstract

[Figure]

We will shorten the abstract and omit a couple of sentences on the data acquisition works.

2) integrate the geology chapter adding at least a few of information and references (if available) about the known formations and geological (regional) evolution of the study area. More important, the structural information require integration on the light of the final interpretation proposed by authors. We can add additional references on the regional geology of Bergslagen. Published articles on the geology of the area are not so many.

Unfortunately, the same on the structural data. Structural data are not so much available on most geological maps produced in Sweden mainly due to the lack of outcrop observations. Only at places where mining is active and borehole data available these are better present. Structural data from our study area are mainly limited to topographic and magnetic lineaments.

Please add at least few sentences and references on the actual stress regime and therefore describe which type of regional (master faults) are mapped in the study area (the main lineaments can be added in Fig.1a as well). This info is important to aid readers not expert on the study region and guide them to the final interpretation.

We will show major structural lineaments from the study area. Stress is even more difficult to present and we think this will be mis-leading! None of the interpreted faults are likely related to the present stress, present stress is mainly dominated by the post-glacial rebound and far-stress from the Atlantic (the ridge). We have no measurement in this area about the state of the stress nor our intention is to say any of the faults interpreted from the seismics and topographic data are recent. They are likely all around 1.9-1.8 Ga.

3) the interpretation should be improved, particularly when introducing the figures 10 to 14 which should be mentioned sequentially in the text.

Figures 10-14 are sequentially mentioned in the text. Figure 10 is brought up in section 6.2 Unmigrated stacked section and Figures 11-13 in section 6.3 where 3D image of the mineralization and potential resources are covered. Figure 14 comes a bit later. We though this way of presenting would help the reader follow both the procedure and interpretation better and avoid going back and forth to these figures.

4) Figs. 12-13-14 can be reduced to three instead of four, and increased in size. Please consider using this scheme: a) only short labels (M1, F1 etc..) b) interpretation labels (such as "Mineralization" etc..) c) only the interpreted model without seismic data

We agree with this comments and revised figures of 3 sets will be presented following this scheme.

Figures and Tables:

We will follow most of the recommendations for reorganizing the figures and their representations such as avoiding white arrows etc. However as for Figures 10 and 11, this way of presenting is better because it honours true orientation of the cure and easier to follow features in their proper orientations. We prefer to keep them as they are. We unsure adding an amplitude bar would really help as the processing work did not honour true amplitude of the data.

Figures 12-14 will be reduced to 3 sets (see above).

We will also make sure parameters of the processing are consistent with those of the table and mentioned text.

F1 appears to have a totally normal movement but we are unsure at what stage this was normal. Most faults in Bergslagen are multiphase and have gone through several stages of compression and extension. Adding a normal fault is very appealing but may mislead. The Blötberget however is likely a "low land" due to this (lost likely) latest normal faulting! We can elaborate slightly on this in the revised table 3.

---

## Author Response (AR1)

**Revision note:**

We provide the original comments in black and our response to them in blue. We thank all the 3 reviewers and editor for finding the manuscript useful for the journal.

**Anonymous Referee #1**

The manuscript shows the 3D seismic data acquisition for mining exploration in Sweden. Although, this survey is not a conventional 3D dataset similar to those acquired in Canada which would provide the opportunity to apply more advanced processing steps (e.g., DMO or PSTM), the authors were able to address the narrow azimuth nature of this survey and showed reasonable results.

We thank the reviewer for the positive comments and favorable review. We are aware of most 3D seismic surveys conducted in Canada for deep mineral exploration purposes. Examples from the Bathurst Mining Camp (such as Brunswick No. 6 and Halfmile Lake), those in Sudbury and recent ones for uranium exploration. They all nearly use the so-called regular geometry setup. Nevertheless, as reported by Cheraghi et al. (2013), even some of these datasets are narrow azimuth and have very irregular offset-azimuth distributions with strong acquisition footprints. We will add some of these references and elaborate further the complexity of narrow azimuth and sparse data. C1 SED Interactive comment Printer-friendly version Discussion paper Regardless of the geology and intended planned azimuth, we anticipate some acquisition footprint in the results however this may not be so obvious as the target is strongly reflective and this may like visual observation of the footprint difficult. Future surveys should likely investigate this further and also if possible a regular 3d seismic dataset to be acquired and compared. This will likely make the Blötberget seismic datasets complete and unique for extensive comparative works and studies. The revised article will address these issues and cite a few more article relevant to this study. Thanks again.

**Anonymous Referee #2**

This manuscript is well written, it can be accepted for publications only subject to minor

revisions (1) The abstract should be prepared more succinctly, in particular, it is unnecessary to introduce the parameters for seismic data acquisition in so many details.

We appreciate the positive and favourable review and comments of the reviewer. They are all to the point and will be addressed in our revised manuscript. We here detail our response to the particular questions raised by the reviewer.

We have shortened the abstract and removed some of the acquisition details.

(2) Line 25. What do you indicate for 10 Mt additional resources? Iron? How did you estimate this potential source? Please indicate the grade of this ore deposit used in the estimate.

Sorry if this was unclear; the ca. 10 Mt estimated additional resources from the seismic data/volume, is based on a lateral and vertical extent of 300 m (300 m by 300 m) and a thickness of 30 m. We have the lateral and depth extent controls from the seismic volume however the thickness is unresolved and is only based on an assumption that the sheet-like deposits have this thickness where drilled through at other locations. The density was assumed around 4000 kg.m3, a reasonable density for economic Fe-content rocks (40-50% Fe). Obviously, this is then is not Fe content rather tonnage of the rocks worth mining. We have further detailed this in the revised manuscript.

(3) To be outspoken, I did not see significant improvements of 3D survey compared to 2D survey in Figure 14. Could you please explain this improvement better?

Well-noted and our bad! Figure 14 is only showing the earlier 2D section but with features/surfaces extracted from the 3D volume. Our intention was to show how the 3D cube helped to identify features that are not in the plane of the 2D section. We have reworked the figure and cation and text to clarify this. Sorry and thanks for noting this.

**Response to the referee#3 comments**:

We thank the positive and favourable review and comments of the reviewer and that the topic is considered "hot" these days. We agree with the reviewer and this is why we wanted to showcase how-to and example works in this current manuscript. We detail rather the most important comments and how they are addressed in the revised manuscript.

General comments: 1) shorten the abstract C1 SED Interactive comment Printer-friendly version Discussion paper

We have shortened the abstract and omitted a couple of sentences on the data acquisition works.

2) integrate the geology chapter adding at least a few of information and references (if available) about the known formations and geological (regional) evolution of the study area. More important, the structural information require integration on the light of the final interpretation proposed by authors.

We have added additional references on the regional geology of Bergslagen. Published articles on the geology of the area are not so many. Unfortunately, the same on the structural data. Structural data are not so much available on most geological maps produced in Sweden mainly due to the lack of outcrop observations. Only at places where mining is active and borehole data available these are better present. Structural data from our study area are mainly limited to topographic and magnetic lineaments. Figure 1 has been changed to add some of this information.

Please add at least few sentences and references on the actual stress regime and therefore describe which type of regional (master faults) are mapped in the study area (the main lineaments can be added in Fig.1a as well). This info is important to aid readers not expert on the study region and guide them to the final interpretation. 3) the interpretation should be improved, particularly when introducing the figures 10 to 14 which should be mentioned sequentially in the text.

We show major structural lineaments from the study area (revised Figure1a). Stress is more difficult to present and we think this will be mis-leading! None of the interpreted faults are likely related to the present stress, present stress is mainly dominated by the postglacial rebound and far-stress from the Atlantic (the ridge). We have no measurement in this area about the state of the stress nor our intention is to say any of the faults interpreted from the seismics and topographic data are recent. They are likely all around 1.9-1.8 Ga.

Figures 10-14 are sequentially mentioned in the text. Figure 10 is brought up in section 6.2 Unmigrated stacked section and Figures 11-13 in section 6.3 where 3D image of the mineralization and potential resources are covered. Figure 14 comes a bit later.

We thought this way of presenting would help the reader follow both the procedure and interpretation better and avoid going back and forth to these figures. The figures and text have been modified as suggested. There should be less going back and forth in the text.

4) Figs. 12-13-14 can be reduced to three instead of four, and increased in size. Please consider using this scheme: a) only short labels (M1, F1 etc..) b) interpretation labels (such as "Mineralization" etc..) c) only the interpreted model without seismic data

We agree with this comments and revised figures to 3 sets following this scheme.

Figures and Tables:

We followed all the recommendations for reorganizing the figures and their representations such as avoiding white arrows etc. However as for Figures 10 and 11, this way of presenting is better because it honours true orientation of the cure and easier to follow features in their proper orientations. We prefer to keep them as they are.

We are unsure adding an amplitude bar would really help as the processing work did not honour true amplitude of the data. We ignored this comment.

Figures 12-14 are reduced to 3 sets (see above). We also fixed inconsistencies of parameters of the processing with those of the table and text.

F1 appears to have a totally normal movement but we are unsure at what stage this was normal. Most faults in Bergslagen are multiphase and have gone through several stages of compression and extension. Adding a normal fault is very appealing but may mislead. The Blötberget however is likely a "low land" due to this latest normal faulting! We have elaborated slightly more on this in the revised table 3 and the main text.